# Mechanical Performance of Single-Graded Copolymer-Modified Pervious Concrete in a Corrosive Environment

**DOI:** 10.3390/ma14237304

**Published:** 2021-11-29

**Authors:** Chaohua Zhao, Xiaoyao Jia, Zhijian Yi, Hualin Li, Yi Peng

**Affiliations:** 1School of Civil Engineering, Chongqing Jiaotong University, Chongqing 400074, China; xiaoyaojxy@126.com (X.J.); yizj63@126.com (Z.Y.); hualin_li199508@163.com (H.L.); 2College of Traffic & Transportation, Chongqing Jiaotong University, Chongqing 400074, China; py.peng@outlook.com

**Keywords:** polymer, copolymer-modified cement concrete, mechanical performance, durability, microstructure

## Abstract

Polymer-modified cement has found numerous applications due to its excellent tensile strength. When cement was modified with a single polymer, its tensile strength and performance marginally increased. However, when a small amount of the flexible polymer latex was added to cement, the mechanical performance of cement considerably improved. In this study, single-graded copolymer-modified pervious concrete was prepared by mixing styrene-butadiene rubber (SBR) with different acrylate polymers, and its mechanical performance and durability were systematically studied through mechanical tests and theoretical analyses. The main findings are as follows: (1) When a waterborne emulsion was added, the freeze resistance of cement concrete increased, and its mass loss rate decreased. Cement concrete with two latexes had higher sulphate corrosion resistance and substantially lower dry shrinkage strain than ordinary cement concrete. (2) Through scanning electron microscopy, the microstructural changes in the cement binder, ordinary polymer-modified concrete, and the copolymer-modified cement concrete used in this study were observed, and the findings were compared with those reported in the literature. (3) Single-graded copolymer-modified pervious concrete exhibited excellent shrinkage performance. This study showed that single-graded copolymer-modified pervious concrete satisfied the performance requirements for use as a paving material for special cases.

## 1. Introduction

Various methods have been developed to improve the functional performance of concrete. With the development of polymer material science, polymer latex has been proven to reduce its brittleness [1,2,3,4]. Polymer latex is added to concrete to obtain polymer concrete (PLC), polymer-modified concrete or polymer cement concrete (PMC or PCC), and polymer-impregnated concrete (PIC) [5,6,7]. In PLC, polymer, instead of cement, is added as a binder and mixed with an aggregate. Portland cement may sometimes be utilised as a filler or fine aggregate. PLC is formed after the mixture is poured, cured, and polymerised. PMC is a composite material obtained by combining cement and aggregate with a polymer material dispersed or dispersible in water. PIC is a composite material containing hydrated cement concrete impregnated with monomers and is obtained after the monomers are polymerised in concrete. Despite the lower modification effect of PMC than that of PIC, researchers in building materials science have become more and more interested in the research of polymer modified cement concrete in recent years because of its simple production and conservative application [1,2,3,4,5,6,7,8,9,10,11,12,13,14]. In addition, considerable research has been conducted to study the physical and mechanical performance of PMC; therefore, it has found several applications.

In addition to the polymer, PMC contains cement paste, mortar, and concrete [8]. PMC and polymer-modified cement paste have found numerous applications because of their better performance than the performance of ordinary concrete and ordinary cement paste for flexural strength, shrinkage, and corrosion resistance. As a rule, polymer-modified cement paste can be utilised as a walling material and is considered an ideal patching material. It can also be used as a bricking, plastering, and waterproofing material. PMC can be used in road construction due to its excellent water resistance, flexural strength, and compressive strength. When utilised as a road surface material, PMC increases the life of the road surface with its good cracking resistance. When used in the construction of bridge decks, PMC and polymer-modified cement paste allow engineers to avoid some complicated construction techniques (e.g., bonding and waterproofing) and reduce engineering costs. Polymer-cement-based composite is a new material having organo-mineral interactions and is characterised by excellent tensile strength, viscosity, dry shrinkage resistance, corrosion resistance, and durability [5,6,7,9]. With these properties, this material has garnered considerable research interest worldwide.

When PMC is prepared using a single polymer, such as rigid styrene-acrylic latex, it retains its high strength and shows a marginal improvement in compressive strength and tensile strength However, even if a small amount of flexible polymer latex is used to prepare PMC, its mechanical performance improves considerably. Such polymer-modified cement has an aggregate structure and shows excellent mechanical performance improvement in relation to the performance of the original components because of an agglutination reaction of two polymer latexes with cement, which is similar to the reaction occurring in metal alloys [10,11]. Therefore, in this paper, the material prepared by completely mixing a copolymer with cement is referred to as a copolymer-modified cement binder. This resultant product is characterised by high toughness and tensile strength. In this study, single-graded copolymer-modified cement concrete was prepared by completely mixing a single-graded aggregate, cement, and copolymer-modified latex to produce a new paving material with excellent water permeability, noise reduction capacity, and mechanical performance [12,13,14].

However, Sofi et al. [15] reported that the incorporation of rubber particles adversely affects the compressive and flexural strengths of concrete due to poor bonding between the cement paste and rubber particles and the softness of the rubber particles. Hence, many researchers have addressed these problems by improving the performance of polymer additives [16,17]. A copolymer was reported to effectively modify pervious concrete, wherein the porosity was increased, and a significant improvement in the later-stage mechanical strength was observed [18]. Benyahia et al. [19] found that emulsion copolymerisation is a convenient, fast, and low-cost alternative to produce the copolymer by dispersing monomer droplets in an aqueous phase. Łaźniewska-Piekarczyk et al. [20] reported that the combination of two different polycarboxyl ethers through emulsion copolymerisation can considerably affect the properties of fresh and hardened high-performance self-compacting concrete. The relationships between pressure methods of measuring air content in fresh concrete and measurements of air-void system parameters using the Air Void Analyzer (AVA) for fresh self-compacting concrete (SCC) and high-performance self-compacting concrete (HPSCC) made with two different superplasticizer (SP) types, air-entraining admixture (AEA), viscosity modifying admixture (VMA) and anti-foaming admixture (AFA) is examined in the paper. The results indicated that the admixtures significantly affect the porosity of SCC and HPSCC. The research results showed that the fluidity of fresh concrete mixture significantly affects the reliability of the measurement using AVA. Lang Lei et al. and Zhang Peng et al. [21,22] have shown that the copolymer can substantially improve the unconfined compressive strength of concrete and strength development of cement-solidified dredged sludge. In addition, acrylate polymers were proven to endow the grout with less toxicity, adjustable gelation time, low permeability, and high compressive strength, thus providing numerous applications of the acrylic grout in various fields, including dams, foundations, and tunnels [23]. Furthermore, acrylate was copolymerised with other materials, such as acrylamide or styrene, to enhance the mechanical, physical, and rheological properties of the grout [24]. However, limited studies have considered improving the mechanical properties of the pervious concrete mixture without sand, and research on the enhancement of acrylate copolymer to PMPC is lacking. In our previous study, we investigated the tensile strength of single-graded copolymer-modified cement concrete by considering it to be similar to a polymer alloy [25]. In this study, we explored the mechanical behaviour of single-graded copolymer-modified cement concrete in a corrosive environment.(1)The mechanical performance and durability indexes, such as freeze resistance, sulphate corrosion resistance, and dry shrinkage performance, of concrete samples with different amounts of the polymer latex were measured.(2)The microstructure of polymer-blended concrete samples was observed using scanning electron microscopy (SEM), and the effect of the microstructure on cement performance was studied.

The findings of this study showed that single-graded copolymer-modified cement concrete can play an important role in structural materials due to its resistance to permeation, cracking, corrosion, and shaking. Consequently, it can find extensive applications in the construction of roads and bridges, airport runways, and marine structures.

## 2. Materials and Methods

### 2.1. Raw Materials and Parameters

In this study, cement, gravel, and compound polymer latex were used. The sample used in this paper is the same as that used in our previous study [25].

(1) Cement: Cement is the most important binding material in concrete. Ordinary silicate cement is utilised in polymer-modified mortar and PMC. In China, road cement is required to meet the technical specifications listed in the Chinese standard JTG D40-2011 [26], according to which road cement should achieve quick hardening and early strengthening to enable rapid resumption of road traffic. Therefore, the selected raw material was P.O 42.5R ordinary silicate cement (Huaxin Cement Co., Ltd., Chongqing, China). The chemical composition of this cement is listed in Table 1.

(2) Coarse aggregate: The selected raw material was stone provided by Jiangchuan Building Material Factory, Chongqing, China. After treating the raw material using several processes—such as screening in a screener, rotating and cleaning in a mixer, and drying in a dry box—single-graded stone with a grain size of 4.75–9.5 mm was obtained.

(3) Polymer: The copolymer latex was prepared by blending styrene-butadiene rubber (SBR) and the acrylate polymer, and the modification effect of a single polymer latex SBR on cement was studied at the early stage [27]. In a previous study [25], different acrylate polymers obtained through various techniques exhibited varying performances. On the basis of a qualitative comparative analysis conducted in that study, we analysed the copolymers of SBR and three typical types of acrylate polymers—XG–6161 acrylate (hereafter denoted as P1), pure acrylate (hereafter denoted as P2), and XG–2135 acrylate (hereafter denoted as P3). The main characteristics of the polymers are listed in Table 2.

(4) Water: The selected raw material was ordinary tap water.

### 2.2. Mixture Proportions for Various Tests

The concrete mixture used contained water, cement, and aggregate in a ratio of 0.41:1:4.7 by weight, based on our previous study [12,13,14]. Various combinations of polymers were tested to identify the combination that improves the mechanical performance of the concrete. P1–P3 were separately mixed with SBR to investigate the effects of copolymers on the performance and properties of concrete. In addition, water was replaced with the copolymer latex. Further, three amounts of each acrylate polymer (i.e., P1, P2, and P3)—10%, 20%, and 30%—were used in the latex, resulting in 11 combinations of the copolymer latex. The mixing proportions of SBR and acrylate polymer for the 11 combinations are listed in Table 3; the control mixture is called Mix ID. The latex-to-cement and aggregate-to-paste ratios of the samples were kept constant at 0.41 and 3.33, respectively.

### 2.3. Sample Moulding and Curing

All the samples were prepared according to the Chinese Standard GB/T 50081-2019 [28], demoulded after 24 h, and placed in a fog room at (20 ± 2) °C and 95% relative humidity for 28 days. The sample moulding process is as follows:

The selected latexes were mixed with the base polymer latex according to the predetermined proportions, and the compound latex was well stirred and completely blended. Cement and stones were weighted according to the predetermined proportions and stirred with an SWJ-60 double-horizontal-shaft forced concrete mixer (Zhengzhou Changcheng Machine Manufacturing Co., Ltd., Zhengzhou, China). The dry mixing time was 2 min. After cement and stones were properly mixed, the completely blended compound latex was slowly added, and the new mixture was further stirred to obtain well-stirred copolymer-modified concrete. The stirring time was 120 s. The stirring process is shown in Figure 1.

Due to the large porosity and loose structure of single-graded copolymer-modified cement concrete, if a vibrator is used, cement paste and stone can be separated. Therefore, a handheld drill rod (Shandong Yikuang Technology Co., Ltd., Shandong, China) was used to tamp concrete. In the sample modelling process, concrete pouring was done in two steps. First, half of the concrete was poured and tamped with the drill rod. Then, the other half was poured, and all concrete was further tamped to ensure the overall compactness and uniformity of the samples. Finally, the surface of the samples was smoothed.

In the sample preparation process, the standard curing method was adopted, followed by plastic film covering. This method was adopted to avoid water loss in the curing process to enable the participation of both polymer and water of the compound latex in the hydration process. This enables the proper formation of polymer cement paste and helps avoid rapid water evaporation, which can lead to incomplete cement hydration and thus adversely affect the concrete strength. Therefore, after the completion of moulding, all the samples were placed in a curing room (Tianjin Huida Experimental Instrument Co., Ltd., Tianjin, China).(temperature: 20 °C± 3 °C; humidity: >90%). The samples were demoulded after curing them for 24 h. They were then covered with a plastic film before further curing at room temperature. The sample dimensions are listed in Table 4.

## 3. Testing Methods

### 3.1. Dry Shrinkage Test

Dry shrinkage is structural shrinkage with drying, which is caused by gradual water evaporation during the hydration or hardening of cement concrete, carbonisation resulting from the chemical reactions of hydration products (calcium hydroxide, calcium silico-aluminate hydrate, and calcium aluminate sulphate hydrate) with airborne carbon dioxide at certain humidity levels, and temperature changes.

The method for the dry shrinkage test was designed in view of the Chinese *Standard for Test Methods of Long-Term Performance and Durability of Ordinary Concrete* [29] (GB/T 50082-2009). The samples are shown in Figure 2. Overall, 17 groups, each with 3 samples, were used. The samples were placed in a standard curing room (temperature: 20 °C ± 3 °C; relative humidity: >90%) and demoulded after curing them for 24 ± 2 h. The samples were numbered, with measurement directions marked. After the completion of demoulding, the samples were measured in the marked directions within 30 min and 3 d, 7 d, 14 d, and 28 d for obtaining the dry shrinkage value. They were then placed in an air-conditioned room (temperature: 25 °C ± 2 °C; relative humidity: <60%) for further curing. The shrinkage values of the samples were measured using a JH-320 alkali aggregate length comparator (Beijing Zhongjian road Instrument Equipment Co., Ltd., Beijing, China) (Figure 3). The samples were analysed using a dial gauge (Shanghai Gaozhi Precision Instrument Co., Ltd., Shanghai, China)with a standard bar length of 295 mm and a measurement accuracy of 0.01 mm. The specific length and dial gauge were calibrated with a standard bar (Shanghai Gaozhi Precision Instrument Co., Ltd., Shanghai, China) before and after each measurement. The formula for calculating dry shrinkage is as follows:S_t_ = (L_0_ − L_T_)/250(1)
where S_t_ is the dry shrinkage rate at a certain age (accuracy: 0.001%), L_0_ is the initially read value (mm), L_t_ is the measured value at a certain age (mm), and 250 is the effective length of the samples.

For each group, the arithmetical mean value of shrinkage for the three samples was taken as the final dry shrinkage value. If the dry shrinkage value for a sample deviated from the mean value by >15%, the mean value of other samples was taken as the final value. If the dry shrinkage values of two samples deviated from the mean value by >15%, the dry shrinkage test was repeated.

### 3.2. Freeze–Thaw Cycle Test

The method for the freeze–thaw cycle test was designed by referring to the slow freezing method outlined in *Standard for Test Methods of Long-Term Performance and Durability of Ordinary Concrete* (GB/T 50082-2009) [29]. The sample dimension was 100 mm × 100 mm × 100 mm. Each group had three samples. The test process is as follows:
(1)The samples were cured in the standard curing room for 24 days and then soaked in water at 20 °C ± 2 °C for 4 days. In the soaking process, the water surface was kept 20 mm higher than the top surface of the sample. The freeze–thaw cycle test was started when the sample age reached 28 days.(2)The samples (Figure 4) were removed when the sample age reached 28 days and wiped dry to remove any surface water. After measuring their dimensions, the samples were weighed and numbered.(3)The samples were inspected at regular intervals. For each group, the freeze–thaw cycle test was stopped when the mass loss reached 5%.(4)The samples were taken out after 50 freeze–thaw cycles and wiped dry to remove any surface water. Their compression strength was then measured.

### 3.3. Sulphate Corrosion Test

Sulphate corrosion is a cyclic occurrence of internal swelling caused by sulphate solution corrosion and internal shrinkage resulting from water content loss during high-temperature drying in cement under the alternating conditions of high temperature and sulphate solution. The cyclic swelling–shrinking phenomenon persists till the alternating high temperature–sulphate solution environment exists, and internal cracks form in cement when its ultimate stress is exceeded.

Characteristics of sulphate attack on concrete: the concrete surface turns white, all edges and corners begin to be damaged, cracks appear, and then concrete peeling occurs. The concrete becomes loose and fragile from compactness. The essence of sulphate erosion damage to concrete is that when the concrete is in the sulphate rich area, sulphate ions enter the concrete and react with the components in cement stone to form refractory salt minerals. These substances can form easily expandable chemical substances such as ettringite and gypsum. The expansion of ettringite and gypsum will cause micro cracks in the concrete, leading to cracking, peeling and the disintegration of concrete [29].

The test method for the sulphate corrosion test was designed by referring to *Standard for Test Methods of Long-Term Performance and Durability of Ordinary Concrete* [30]. An actual industrial contamination environment was considered the background to simulate the sulphate corrosion resistance of copolymer-modified cement concrete and plain concrete in a natural sulphate corrosion situation.

To study the impact of sulphate corrosion on the durability of ordinary cement concrete and single-polymer-modified cement concrete with different amounts of the polymer latex, the original components of cement should be free from sulphate, thereby ensuring that any sulphate corrosion is external. The test had 17 groups, each with 6 samples, or 102 samples in total. The samples, designed as 100-mm-edge cubes, were divided into seven types. The samples were cured in the standard curing room for 28 days, soaked in clear water and 5% NaSO_4_ solution for 14 and 28 days, respectively, and then observed for surface damage and their compression strength was measured, on the basis of which the corrosion coefficient of compression strength of cement was calculated. The formula for the coefficient is as follows:K_f_ = (f_0_ − f_n_)/f_0_(2)
where K_f_ is the corrosion coefficient of compression strength (%), f_0_ is the mean value of the compression strength of the samples without sulphate corrosion (MPa), and f_n_ is the mean value of the compression strength of the samples with sulphate corrosion (MPa).

### 3.4. SEM Test

To observe the modification of concrete caused by the copolymers, the morphologies, microstructural characteristics of samples are observed by Zeiss Gemini Sigma 300 VP SEM machine (Carl Zeiss AG, Jena City, Germany). The working distance of the machine’s lens and samples is 8.5 mm and the general test steps are conducted according to study [31].

There are three types of samples: ordinary cement concrete, single-polymer-modified cement concrete, and copolymer-modified cement concrete. Single-polymer-modified cement concrete contains only one type of polymer, i.e., 100% latex SBR. Copolymer-modified cement concretes that contains three type of polymers, respectively, i.e., 70% latex SBR + 30% latex P1, 70% latex SBR + 30% latex P2 and 70% latex SBR + 30% latex P3 were selected for the analysis.

Sample preparation: first, five disposable paper cups are used and labelled for this test corresponding to the test piece one by one. Second, samples are taken during the 28-day compressive strength test of cement-based materials. A large number of small fragments are generated during the compressive test. The size of the sample selected from the fragments shall not exceed 3.5 mm * 3.5 mm. The sample surface shall be flat for observation during scanning. Third, dry the sample in an electric blast drying oven (Shanghai Keheng Industrial Development Co., Ltd., Shanghai, China) at 60 °C for 24 h. Fourth, put the dried specimen in the gold plating instrument (Foshan Foxin Vacuum Technology Co., Ltd., Foshan, China) for gold plating. Gold plating can improve the conductivity of the specimen and improve the clarity of the scanned image of the specimen. The experiment was conducted in the materials laboratory of Chongqing Jiaotong University. The scanning process is shown in Figure 5.

## 4. Results and Discussion

### 4.1. Dry Shrinkage Test

#### 4.1.1. Test Results

The results of the dry shrinkage test for the various groups of cement paste samples at various ages from day 1 to 28 are listed in Table 5.

#### 4.1.2. Analysis of Test Results

The following conclusions can be derived from the strain values given in Table 5:
(1)For all groups of the copolymer-modified cement concrete samples, the dry shrinkage rate first gradually increases and then stabilises when the sample age increases. Before 7 days, the dry shrinkage rate increases at the fastest rate, reaching approximately 85% of the dry shrinkage rate on the 28th day. After 7 days, the dry shrinkage rate increases at a steady rate.(2)After the polymer latex is added, the dry shrinkage performance of the cement paste samples substantially improves with the latex content. After P1 and P2 are added, the dry shrinkage rates of the copolymer-modified cement paste are decreased by 6.68–12.76% and 7.78–12.21%, respectively. By contrast, after P3 is added, the dry shrinkage rate is decreased by only 1.23% and increased by 1.23% when the latex content is 30%. These findings indicated that the dry shrinkage performance of copolymer with SBR and P1 or P2 is better than that of the copolymer with SBR. Furthermore, the dry shrinkage performance of the copolymer with SBR and P3 decreases when the P3 content increases.

The main reasons for the improvement in the shrinkage performance of polymer-modified cement are as follows: Protective films are generated by the polymer on the surface of the hydrated cement products. These films prevent internal water evaporation and promote cement hydration reaction. Furthermore, solid components in the polymer become dispersed in the microcracks of the cement stone, which increases cement stone compactness and reduces the number of water evaporation paths in the cement stone. In a dry environment, the water pressure in the pores in the cement stone is maintained, which reduces the dry shrinkage rate of the cement paste.

### 4.2. Freeze–Thaw Cycle Test

#### 4.2.1. Analysis of External Damage

The results of this test showed the compression failure of the samples after the freeze–thaw cycles as gradual damage starting from the sample surface. The damage process of the samples can be generally divided into three parts as shown in Figure 6:
(1)The surface of the cubic samples of the copolymer-modified cement binder is smooth and damage-free.(2)As the freeze–thaw cycle test proceeds, the sample colour changes from grey to white, and many tiny pores gradually form on the sample surface.(3)As latex loss in the samples due to water soaking continues, the surface pit area and the number of pores increase in the samples, and extensive arc-shaped fall-off occurs at the edges and corners in some samples.

#### 4.2.2. Compression Strength Loss and Mass Loss after Freeze–Thaw Cycles

The following conclusions can be derived from the test data given in Table 6:
(1)After 50 freeze–thaw cycles, a minor mass loss occurs in all the samples, and damage appears on the sample surface to varying degrees. The test results indicated that the freeze resistance performance of cement binders with various polymer combinations is significantly better than that of the ordinary cement paste.(2)The mass loss rates of the copolymer-modified cement binder are lowest for the latex content of 30% P1 and 30% P2, which show the mass loss rates of 0.04% and 0.03%, respectively, or 7 and >9 times that of ordinary cement binder, respectively. The freeze–thaw failure develops from the sample surface and travels inside. Due to the restrictions of test conditions, only 50 freeze–thaw cycles were conducted; consequently, freeze–thaw failure from the surface to the inside of the samples could not be observed, and only edge and corner damages and arc-shaped damages were detected in some samples.

### 4.3. Sulphate Corrosion Test

The following conclusions can be derived from the values given in Table 7:
(1)The corrosion coefficients of the compression strength of various groups of the copolymer-modified concrete samples are smaller than those of ordinary skeleton-pore cement concrete and ordinary polymer-modified skeleton-pore cement concrete to varying degrees. Among the samples, the corrosion coefficient of compression strength of ordinary skeleton-pore cement concrete is the highest, reaching 45.32%, and its sulphate corrosion resistance is the lowest.(2)After the latex is added, the sulphate corrosion resistance performance of the samples improves. Furthermore, the resistance performance improvement increases with the latex content. The sulphate corrosion resistance performance of copolymer-modified cement concrete blended with latex P3 is the optimum, which is higher than that of ordinary cement concrete by approximately 26.14%. However, its after-test compression strength is still lower than that of other types of copolymer-modified cement concrete because of its low before-test compression strength.(3)In comparison, the sulphate corrosion resistance performance of copolymer-modified concretes blended with latexes P1 and P2 is optimum, and the difference between these is considerably small. Protective films are properly formed by the polymer latex on the surface of the hydration products of cement, thereby effectively reducing the sulphate content in cement, as well as ettringite and gypsum contents, which are materials with expansibility.

### 4.4. SEM Test

The SEM magnification was adjusted to 500 times, and the cross-sectional SEM images of the ordinary cement and polymer-modified cement binders are shown in Figure 7.
(1)As shown in Figure 7a, numerous pores are caused by cement gel particles in ordinary cement binder; hence, its surface is coarse. After the latex is added, such pores are filled by the solid particles in the polymer latex, resulting in lower porosity and smoother surface.(2)As shown in Figure 7b, the polymer latex results in few pores on the surface of cement stones. Furthermore, scattered sheet structures and protective films caused by the latex are formed on and tightly connected with the surface of the cement binder.

The comparative analysis of the SEM images presented in Figure 7 shows that the microstructure of the cement binder changes remarkably when the SBR content decreases and the P1 content increases. Under an SEM magnification factor of 500, the cement binder surface becomes coarse and scattered with tiny pores. This shows that the addition of P1 causes significant changes in the microstructure of polymer cement, which may also attribute to the significant improvement in the properties of polymer cement after blending. However, a more detailed investigation and deeper cause analysis must be studied in the future.

## 5. Conclusions

In this study, the mechanical performance of copolymer-modified cement concrete in a corrosive environment was studied. Several durability performance indexes of various concrete samples were determined by testing parameters, such as freeze resistance, corrosion resistance, and dry shrinkage. The microstructures of the samples were observed through SEM, and the effect on cement performance was analysed.

The main conclusions are as follows:(1)The dry shrinkage coefficient of polymer modified cement is less than that of ordinary cement concrete, and the shrinkage of blended polymer cement is better than that of polymer modified cement. Among them, the blending of P1 and P2 has the best improvement effect.(2)The frost resistance of copolymer-modified cement concrete was analyzed by compressive strength loss and mass loss. The test shows that the compressive strength of the concrete with the addition of polymer emulsion is lower than that of the ordinary cement concrete, but the compressive strength and the mass loss rate are better than those of ordinary cement concrete, that is, its frost resistance is better than that of the ordinary cement concrete.(3)A comparative analysis of the loss rates of compression strength after soaking in 5% sulphate solution for 14 and 28 days showed that copolymer-modified cement concretes with P1 and P2 have the highest sulphate corrosion resistances, and is better than a simple polymer modified cement concrete and ordinary cement concrete.

In this study, the mechanical performance and durability of copolymer-modified cement concrete were studied. The effects of various copolymers on the mechanical performance of concrete at the same polymer-to-cement ratio were analysed. Furthermore, some preliminary conclusions for future research on new paving materials were proposed. However, further study in some areas is required. For example, the durability of materials such as fatigue performance needs to be further studied, and there is still the problem that the amount of polymer latex is too high. It is necessary to further improve the performance of latex to reduce the amount of latex to reduce the cost, so as to be applied to more fields.

## Figures and Tables

**Figure 1 materials-14-07304-f001:**
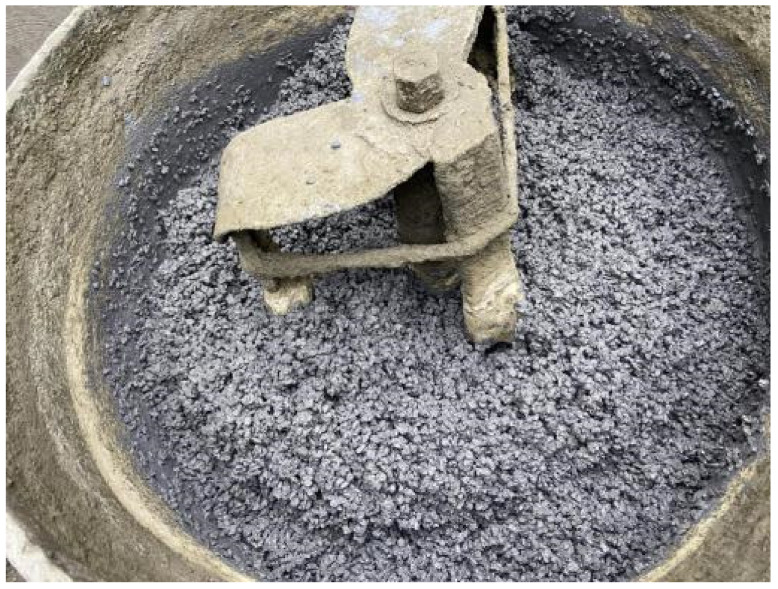
Stirring of copolymer-modified pervious concrete mixture.

**Figure 2 materials-14-07304-f002:**
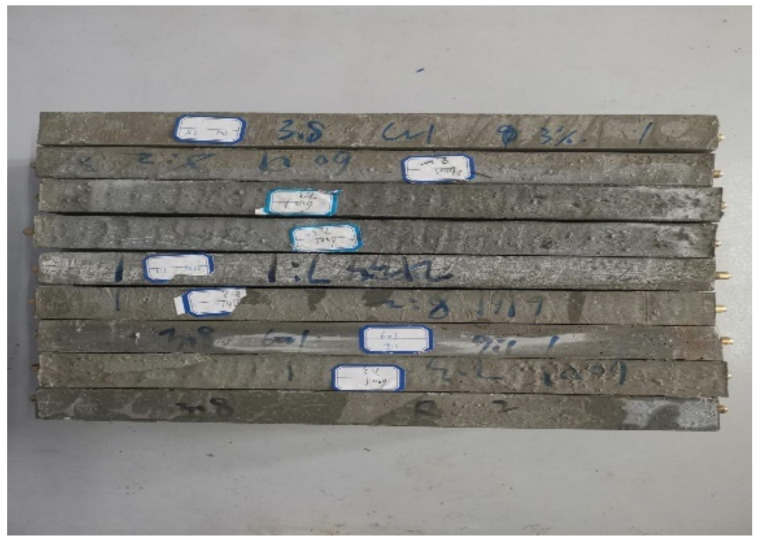
Dry shrinkage test sample.

**Figure 3 materials-14-07304-f003:**
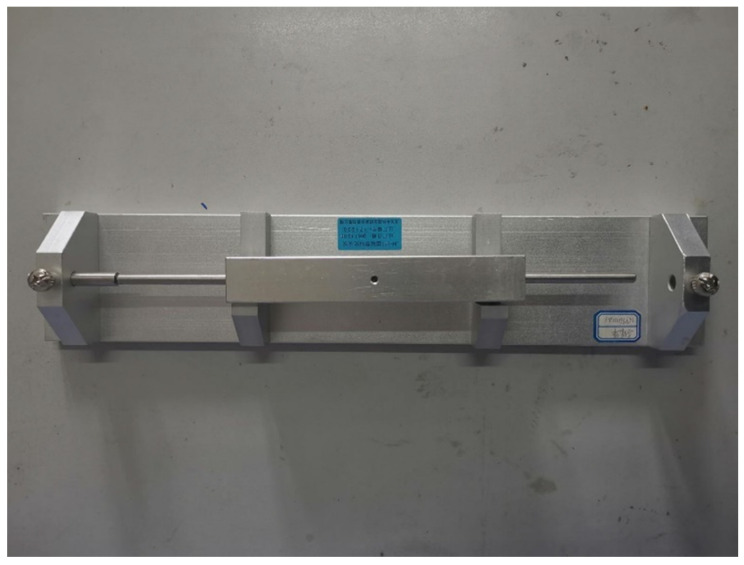
JH-320 alkali aggregate length comparator.

**Figure 4 materials-14-07304-f004:**
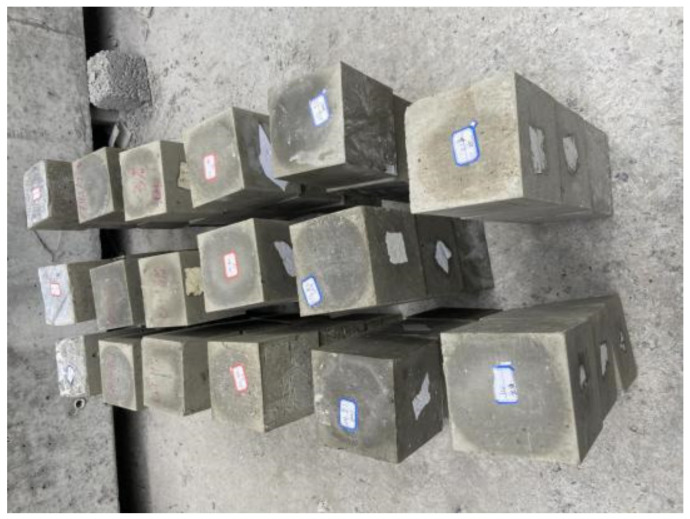
Test specimen for freeze thaw.

**Figure 5 materials-14-07304-f005:**
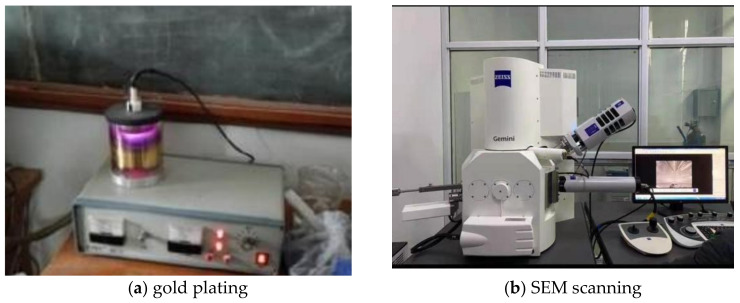
SEM scanning process.

**Figure 6 materials-14-07304-f006:**
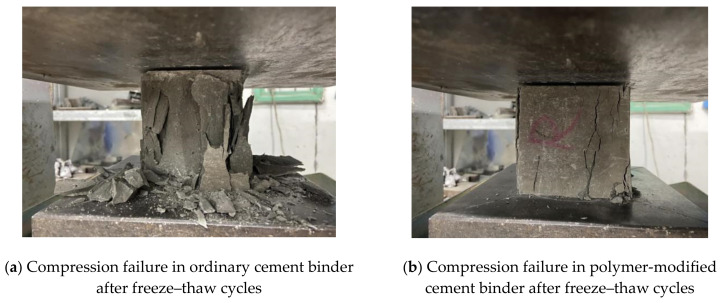
Compression failure patterns of the samples after freeze–thaw cycle tests.

**Figure 7 materials-14-07304-f007:**
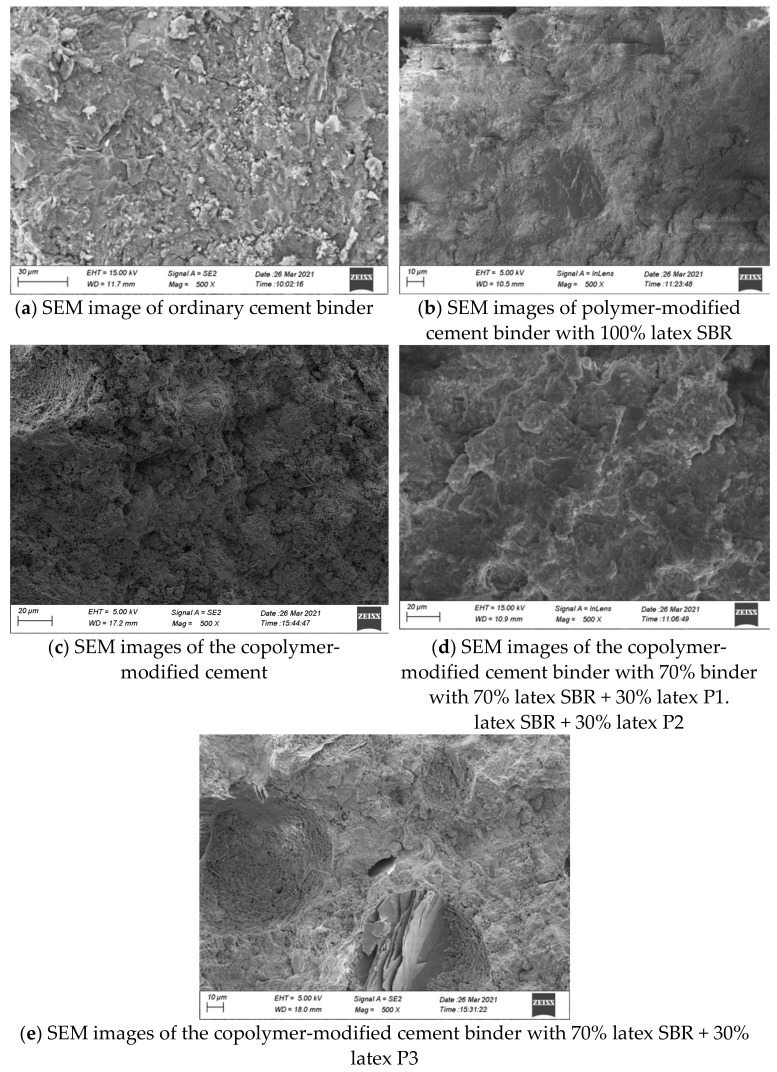
SEM images of the ordinary cement and polymer-modified cement binders.

**Table 1 materials-14-07304-t001:** Chemical composition of cement (%).

Component	SiO_2_	Fe_2_O_3_	Al_2_O_3_	CaO	MgO	SO_3_
Content	22.4	3.15	5.6	59.58	2.58	2.42

**Table 2 materials-14-07304-t002:** Polymer latex characteristics.

	SBR	P1	P2	P3
Chemical family	Styrene-butadiene rubber	XG–6161 acrylate	XG–6001 acrylate	XG–2135 acrylate
Solid content, wt.%	45 ± 1	56 ± 1	55 ± 1	50 ± 1
pH	7 to 8.5	6.5 to 8	7 to 9	7 to 9
Density [kg/m^3^]	1.20 to 1.22	1.08 to 1.10	1.10 to 1.12	1.18 to 1.20
Viscosity [mpa.s]	1500 to 2500	500 to 1500	1000 to 2500	800 to 2500
Appearance	Milky white liquid emulsion	Milky white and greyish liquid emulsion	Milky white and greyish liquid emulsion	Milky white and greyish liquid emulsion

**Table 3 materials-14-07304-t003:** Mixing proportions of copolymer latex.

Control Mixes	Mix ID	Control Mixes	Mix ID
100% Water	W	80% SBR + 20% P2	SP2_20%
100% SBR	S	70% SBR + 30%P2	SP2_30%
90% SBR + 10% P1	SP1_10%	90% SBR + 10% P3	SP3_10%
80% SBR + 20% P1	SP1_20%	80% SBR + 20%P3	SP3_20%
70% SBR + 30% P1	SP1_30%	70% SBR + 30% P3	SP3_30%
90% SBR + 10% P2	SP2_10%	-	-

Note: 100% water indicates pervious concrete without polymer latex, while 100% SBR denotes that without additives. Control mixes for other copolymers are based on the SBR-to-additive ratio (i.e., 90% SBR + 10% P1 indicates that the SBR’s portion in the latex was 90%, while P1 portion was apparently 10%).

**Table 4 materials-14-07304-t004:** Sample dimensions for various tests.

Test Type	Sample Type	Sample Dimensions	Age	Group Number	Sample Number
Dry shrinkage test	Copolymer-modified cement paste	25 mm × 25 mm × 280 mm	1d, 3d, 7d, 14d and 28d	17	51
Freeze–thaw cycle test	Copolymer-modified cement paste	100 mm × 100 mm × 100 mm	28d	17	51
Sulphate corrosion test	Copolymer-modified concrete	100 mm × 100 mm × 100 mm	28d	17	68

**Table 5 materials-14-07304-t005:** Dry shrinkage strain in copolymer-modified cement paste (1 × 10^−6^).

-	1d	3d	7d	14d	28d
Water	222	674	1107	1265	1466
Latex SBR	236	633	1041	1139	1290
SP1_10%	278	646	1055	1137	1209
SP1_20%	266	582	958	1035	1096
SP1_30%	238	501	772	861	905
SP2_10%	255	675	1059	1192	1287
SP2_20%	254	646	1032	1142	1218
SP2_30%	277	624	986	1088	1152
SP3_10%	264	699	1067	1164	1348
SP3_20%	261	687	1009	1136	1304
SP3 30%	273	710	1043	1168	1372

**Table 6 materials-14-07304-t006:** Freeze–thaw test data for various groups of the copolymer-modified cement paste samples.

Compression Strength (MPa)				Mass (g)		
	Before Test	After Test	Loss Rate	Before Test	After Test	Loss Rate
Water	53.33	43.06	19.26%	1920.46	1915.07	0.28%
Latex SBR	46.25	38.75	16.22%	1853.5	1849.87	0.20%
SP1_10%	44.88	41.09	8.44%	1589.54	1588.6	0.06%
SP1_20%	43.54	40.29	7.46%	1647.4	1646.6	0.05%
SP1_30%	41.83	38.56	7.82%	1682.18	1681.5	0.04%
SP2_10%	46.03	40.52	11.97%	1678.83	1677.94	0.05%
SP2_20%	45.66	41.84	8.37%	1791.33	1790.66	0.04%
SP2_30%	44.15	41.15	6.80%	1704.95	1704.41	0.03%
SP3_10%	37.15	31.92	14.08%	1828.18	1826.3	0.10%
SP3_20%	37.33	32.05	14.14%	1814.17	1812.5	0.09%
SP3_30%	38.97	32.96	15.42%	1725.79	1724.4	0.08%

**Table 7 materials-14-07304-t007:** Sulphate corrosion test results.

		Ordinary Compression Strength	Compression Strength under Sulphate Corrosion for 14d	Corrosion Coefficient of Compression Strength	Compression Strength under Sulphate Corrosion for 28d	Corrosion Coefficient of Compression Strength
Water	-	14.9	11.26	24.43%	7.87	47.18%
Latex SBR	-	20.89	16.56	20.73%	12.05	42.32%
Latex P1	SP1_10%	20.25	17.09	15.60%	13.43	33.68%
	SP1_20%	21.86	18.76	14.18%	14.97	31.52%
	SP1_30%	22.53	19.66	12.74%	16.21	28.05%
Latex P2	SP2_10%	20.99	17.43	16.96%	14.61	30.40%
	SP2_20%	22.85	18.73	18.03%	15.53	32.04%
	SP2_30%	23.88	19.89	16.71%	16.76	29.82%
Latex P3	SP3_10%	13.17	11	16.48%	8.71	33.86%
	SP3_20%	13.13	11.04	15.92%	8.48	35.42%
	SP3_30%	13.18	10.85	17.68%	8.22	37.63%

## Data Availability

The data used to support the findings of this study are included within the article.

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
