# Peer review of "Mechanical Performance of Single-Graded Copolymer-Modified Pervious Concrete in a Corrosive Environment"

_materials, 2021, doi:10.3390/ma14237304_

Round 1

Reviewer 1 Report

The paper deals with a potentially interesting topic, though it needs much refinement. Some remarks below.

Line numbering is missing – referencing to a specific part of the document is almost impossible

Introduction must be improved / extended. In the present form, very little information on the recent scientific findings in the area being studied is present.

“This resultant product is characterised by high strength, compressive strength and tensile

strength.” – what do authors mean with the first mentioned “strength”?

Table 1 – the oxides formulas should be written with the numbers in subscript

Table 2 – symbols and units – subscript/superscript, capital letters (for example PH…)

Latex – could authors provide some information regarding molecular weight and its distribution?

3.4 SEM test – how can be the imaging signal generated by absorbed electrons? The whole section must be rewritten, in present form it lacks scientific soundness

Reviewer 2 Report

The manuscript concerns mechanical performance of Single-Graded Copolymer-Modified Pervious Concrete under corrosive environment.

The manuscript is a continuation of the research from an earlier publication in the Materials [15].

The manuscript is interesting, experiments and analyzes well planned and executed. Insightful interpretation of the obtained results.

However, the manuscript contains a number of errors/inaccuracies that need to be corrected/clarified.

The Abstact; First use of SBR and SEM abbreviations, abbreviations should be explained.

The sentence "Cement concrete with two latexes had ..." may be confusing, please specify. Especially when we compare it with another sentence: “3) Polymer: A copolymer latex was prepared by emulsion copolymerisation of styrene-butadiene rubber (SBR) and acrylate polymer [17].”

2.1; (3) Polymer; A paragraph may suggest that other (different) acrylic polymers were used than in [15]. Please specify the description.

2.2 Mixture Proportions for Various Tests, It should be noted that all samples were prepared as in the previous work ([15]).

3.2. Freeze-thaw Cycle Test and 3.3. Sulphate Corrosion Test, I propose to add the standard number in the description (GB/T 50082-2009).

Tables 5, 6 and 7; The samples markings other than before, please compare with the samples markings in the Table 3.

4.3.2. Analysis of test results; what do “Latex A” and “Latex F” mean, these designations no longer appear in the text?

4.4. SEM Test; comparing SEM images is sometimes difficult due to different magnification. can it not be standardized?

Paragraph under Fig.10; The description lacks information with what the images Fig.10 is compared to.

Paragraph under Fig.12; I would suggest, at the beginning of the description, add information that it concerns samples with latex P3.

Reviewer 3 Report

The paper materials-1408867 presents an interesting research experimentally investigating the mechanical behaviour of pervious concrete mixtures subjected to freeze-thaw cycles combined with sulphate attacks.

The topic is interesting and well-aligned with the scope of the journal. The English language and style are good and concepts are expressed in a clear manner. The novelty of the paper is clearly identifiable.

The mechanical tests here presented are supported by other useful analyses, like SEM microscopy and the results are well presented and argued.

My only suggestion concerns the conclusions section, which should be slimmed down and should be rearranged to sketch up the essential knowledge gain from the present study (avoiding to enter the details, that one can see in the body of the manuscript) and a very short overview of the future developments and knowledge gaps.

For the sake of clarity and of an easier comparison, I invite the Authors to gather the SEM images (fig.8 - 13) in a single figure with the labels (a - e).

Reviewer 4 Report

The paper presents the mechanical performance data of a copolymer modified pervious concrete  submitted to freeze-thaw  and sulphate exposure with respect to shrinkage  The English, while of a reasonable standard, does need reviewing to improve the quality of the publication and for ease of understanding.

Specific comments are

PC is generally used to represent Portland cement, while it is defined in this paper as polymer concrete it is suggested that the abbreviation is changed tp PLC to avoid confusion.

References should be added to support statements regarding the popularity of PMC at the end of page 1.

Page 2

amount can be deleted in ‘shrinkage amount’

Change remarkable to good

Change insufficient to lacking

Please provide a summary of the findings for the reference Lazniwska, what is the considerable effect mentioned.

Figure 1 and Figure 2 should either be deleted or labels applied to identify the key information shown in the figures, ie what are the differences being illustrated in the four latex materials as they look very similar and no pertinent information is provided by the images.

Check font/ content Table 1 (ie pH not PH)

Provide rationale as to the selection of 10, 20 and 30% replacement of water by latex

The information of the specific laboratory employed can be deleted, the standard used is sufficient.

Give details of standard curing method employed and the Standard used (if appliucable)

Figure 5 can be deleted as this is common equipment employed for length comparator as can Figure 7 for same reason.

Figure 6 can also be deleted or labels applied to identify the latex additions to each cube to identify those with 10, 20 and 30% additions and the control.

What is meant by cement stone? Does this mean concrete or is it a specific component of the concrete

Why was polishing not employed in the SEM specimens, please give rationale for selection of specimens. Also what is reason for using fragments from compressive tests which could be damaged due to stresses in testing rather than control specimens that were not subject to compressive testing?

Please expand upon the statement that protective films are properly formed on the surface of the hydrated products. What is the exact meaning of properly (ie over the entire hydrated products?) and how does this film reduce the cement content of the cement? Is this unreacted cement or the hydration products formed such as ettringite and gypsum by the reaction of the cement and the sulphate. What is the evidence that the sulphate content has been reduced. Is there less sulphate in P1 and P2 than P3 (eg EXD analysis for ettringite and gypsum or SEM analysis to observe sulphate products and XRF data?

In Figure 9, are these the surface of the fragments from the compressive strength tests, if so are they internal surfaces or the external surface of the specimen?

When authors state that an oily latex does not fully dissolve but scatters the cement binder what is the evidence for this statement (SEM images?)

Where is evidence for microcracks as noted in the conclusions, again is this from SEM analysis, if so these should be identified in the SEM images included in the paper.

Check format of the references 15 and onwards as these appear different to those of 1-14

Round 2

Reviewer 1 Report

I am really sorry, but I do not see much improvement with respect to the first version. Especially the SEM paragraph in the Methods section really must be rewritten. My primary concern is with chapter 3.4, lines 264-282. The chapter looks like a text taken from an introductory textbook on SEM imaging and is thus highly inappropriate in a research article. Besides, the authors do not mention the specifications of the apparatus - producer, type, etc. Were the samples observed "as they were" or with a surface coating? What was the acceleration voltage? How can possibly absorbed electrons (line 268) generate any signal? I urge the authors to use the review mode in MS Word to keep track on what has been changed and let the changes be visible.

Reviewer 4 Report

The revisions have significantly improved the quality of the paper and addressed the comments very well. The following changes are suggested;

Conclusion 4 should be deleted as this does not actually give any specific conclusion stating that the results are well explained is not a conclusion.
